# Overeaters Anonymous: An Overlooked Intervention for Binge Eating Disorder

**DOI:** 10.3390/ijerph18147303

**Published:** 2021-07-08

**Authors:** Brenna Bray, Boris C. Rodríguez-Martín, David A. Wiss, Christine E. Bray, Heather Zwickey

**Affiliations:** 1Helfgott Research Institute, National University of Natural Medicine, Portland, OR 97201, USA; hzwickey@nunm.edu; 2School of Undergraduate and Graduate Studies, National University of Natural Medicine, Portland, OR 97201, USA; 3Proyecto PlaniFive, 28017 Madrid, Spain; borisrodriguez@metriri.com; 4Department of Community Health Sciences, Fielding School of Public Health, University of California Los Angeles, Los Angeles, CA 90095, USA; dwiss@ucla.edu; 5Amherst H. Wilder Foundation, Saint Paul, MN 55104, USA; bray0021@umn.edu

**Keywords:** eating disorder, binge eating disorder, Overeaters Anonymous, twelve steps, spirituality, food addiction

## Abstract

The purpose of this communication is to provide an overview as well as the strengths and weaknesses of Overeaters Anonymous (OA) as an intervention for binge eating disorder treatment. Binge eating disorder is associated with low remission rates, high relapse rates, treatment dissatisfaction, and high rates of failure to receive treatment attributed to stigma, misconceptions, lack of diagnosis, access to care, and inadequate insurance coverage. New interventions are needed that can overcome these barriers. OA is a twelve-step program and established fellowship for individuals who self-identify as having problematic relationships with food or eating. OA can be referred clinically or sought out by an individual confidentially, without a diagnosis, and free of charge. OA’s Nine Tools, Twelve Steps, and Twelve Traditions can provide structure, social support, and open, anonymous sharing that fosters a sense of connection and belonging. This may provide benefit to individuals who value structure and social support in their recovery. The tradition of anonymity may also create some challenges for conducting research and may explain the shortage of empirical support. This commentary reviews existing research findings on the effectiveness of twelve-step interventions and OA. Common misunderstandings about and within OA are also addressed and OA’s limitations are discussed. Overall, OA provides a promising option for binge eating disorder treatment that warrants clinical research on its feasibility and efficacy in a way that respects and protects its tradition of anonymity.

## 1. Introduction

### 1.1. Binge Eating Disorder and the Need for New Treatment Options

Cognitive behavioral therapy is the current standard treatment for binge eating disorder [1] but has mixed outcomes [2] and is associated with low remission rates (52–62%) [3] and 20–60% relapse rates [2] pre-pandemic. Pharmacotherapeutic interventions include SSRI antidepressants (e.g., citalopram, escitalopram, and sertraline), anticonvulsants (specifically topiramate), and anti-obesity medications (e.g., orlistat and sibutramine) [1]. Although pharmacotherapies have mixed results and guidelines, they do not improve cognitive behavioral therapy’s success [1,2]. National survey data collected between 2001–2003 [4] provide some of the most up-to-date information available on binge eating disorder prevalence and treatment nationally [5]. Although this data is outdated and a new survey is warranted, it found that 56.4% of individuals with binge eating disorder never receive or pursue treatment for their illness [4] due to stigma, misconceptions, lack of education, diagnosis, access to care, and inadequate insurance coverage [4,6,7]. Binge eating disorder has also been associated with treatment dissatisfaction and early discontinuation of care [7]. Furthermore, the Coronavirus disease 2019 (COVID-19) and its subsequent lockdown protocol have increased binge eating behaviors in individuals with- and without eating disorders [8,9,10,11], and have further impacted treatment satisfaction and modality [10,12]. The increases in binge eating behaviors seem to have remained post-pandemic [11], emphasizing the need for research identifying new interventions that can overcome barriers and limitations associated with current BED treatment options.

### 1.2. Overeaters Anonymous: An Overlooked Eating Disorder Intervention

Overeaters Anonymous (OA) is a mutual-support, twelve-step intervention and established fellowship for individuals who self-identify as having problematic relationships with food or eating [13,14,15,16]. OA members use the Twelve Steps (Table 1) [14,15,17,18,19], Nine Tools (Table 2) [17,20,21,22], and Twelve Traditions (Table 3) [15] of OA, adapted from Alcoholics Anonymous (AA), to support each other in “recovering from unhealthy relationships with food and body image [16].” OA can be used as an adjunct/component of a multi-disciplinary treatment program [23,24,25] and can also be used independently by an individual or referred as an adjunct or standalone treatment (see Section 2.3) [14,16,26,27]. The OA fellowship currently has >60,000 members with >6000 groups meeting in >75 countries and >20 languages (including ESL) [28,29]. OA meetings are conducted face-to-face and remotely by telephone and online, with >3500 face-to-face meetings, >325 telephone meetings, and >630 online meetings currently listed on the OA website (https://oa.org/find-a-meeting, accessed on 4 November 2020) [28]. OA is without dues or fees, open to all members, and rests on a tradition of participant confidentiality and anonymity [13,14,15,16], thus overcoming many of the barriers that often prevent individuals from accessing treatment (e.g., stigma, lack of diagnosis, access to care, and inadequate insurance coverage) [4,6,7]. 

OA is a non-professional, non-affiliated, non-denominational (non-religious), self-supporting fellowship of individuals who “have recovered” from compulsive eating [19] and whose primary purpose is “to show other [sufferers] *precisely how they recovered*” [19] in order to stay recovered themselves [14,15,16,19]. OA rests on the principles of letting go of self-sufficiency, asking for help, seeking guidance from a power greater than oneself, abstaining from problematic foods and/or eating behaviors that produce craving/loss of control, and sharing the message of recovery with other sufferers in order to stay “abstinent” [13,14,15,16]. Although OA is non-affiliated, non-denominational, and non-religious, it does have a spiritual component, as addressed in its literature [15,18,19], Steps 2–3 [15,18,19] in the Table 1 below, and in Section 2.1, as well as in the supplemental text provided in Appendix A.

Within the OA fellowship, most OA groups, formats, and sponsors suggest that members: (1) abstain from problematic foods and/or eating behaviors; (2) work through OA’s Twelve Steps with a sponsor (Table 1); (3) use the Nine Tools of OA (Table 2) to reinforce one’s own recovery [13,14,15,16,17,22,27,30]. OA defines a sponsor as an OA member who is committed to his or her own abstinence and to “living the Twelve Steps and Twelve Traditions to the best of [his or her] ability [31].” Sponsors work one-on-one with other OA members (“sponsees”) and share their experience, strength, and hope while guiding “sponsees” through the Twelve Steps of the program and reinforcing OA’s Nine Tools, slogans, and other resources [32]. OA encourages newcomers to “find a sponsor who has what [s/he wants] and ask how it was achieved [31].” An individual therefore may choose his/her own sponsor, and may work with more than one sponsor or change sponsors at any time [31]. Sponsors and sponsorship vary significantly within OA, contributing to variation in individual OA recovery (see Section 2.3).

**Table 1 ijerph-18-07303-t001:** The Twelve Steps of Overeaters Anonymous.

Step Number and Description, According to OA’s Twelve Steps [15]		Principle/Virtue [15,33]	Abstinence/Relapse Correlation [27]	Use
Step 1	We admitted that we were powerless over food—that our lives had become unmanageable [15].	Honesty [15,33]	N/A	Steps 1–3: Immediate Emphasis on Physical Abstinence, Behavioral Change, Humility, & Cultivating Faith.
Step 2	Came to believe that a Power greater than ourselves could restore us to sanity [15].	Hope [15,33]	N/A	
Step 3	Made a decision to turn our will and our lives over to the care of God as we understood Him [15] [See Section 2.1 *“Is Overeaters Anonymous a Religoius Program?”*].	Faith [15]/Surrender [33]	N/A	
Step 4	Made a searching and fearless moral inventory of ourselves [15].	Courage [15,33]	A: 0.26, R: 0.3	Steps 4–9: Initially within 1st Year. Emphasis on Emotional & Behavioral Growth. Cultivating Honesty, Openness, & Willingness.
Step 5	Admitted to God, to ourselves, and to another human being the exact nature of our wrongs [15].	Integrity [15,33]	N/A	
Step 6	Were entirely ready to have God remove all of these defects of character [15].	Willingness [15,33]	N/A	
Step 7	Humbly asked him to remove our shortcomings [15].	Humility [15,33]	N/A	
Step 8	Made a list of all persons we had harmed, and became willing to make amends to them all [15].	Self-discipline [15]/Love [33]	N/A	
Step 9	Made direct amends to such people wherever possible, except when to do so would injure them or others [15].	Love [15]/Responsibility [33]	A: 0.26, R: 0.24	
Step 10	Continued to take personal inventory and when we were wrong promptly admitted it [15].	Perseverance [15]/Discipline [33]	N/A	Steps 10–12: Maintenance Steps (Ongoing); Emphasis on Spiritual Growth, Service; Cultivating Altruism.
Step 11	Sought through prayer and meditation to improve our conscious contact with God as we understood Him, praying only for knowledge of His will for us and the power to carry that out [15]. [See Section 2.1 *“Is Overeaters Anonymous a Religoius Program?*”]	Spiritual Awareness [15,33]	A: 0.2, R: 0.16	
Step 12	Having had a spiritual awakening as a result of these steps, we tried to carry this message to other compulsive overeaters, and to practice these principles in all of our affairs [15].	Service [15,33]	N/A	

The Twelve Steps of Overeaters Anonymous (OA) [15], based on AA’s Twelve Steps [18], emphasize acceptance and surrender, honesty and humility, behavioral change, ownership for one’s actions and repercussions, spiritual awakening, and selfless service [14,15]. **Columns 1 and 2** identify each of the Twelve Steps outlined in a twelve-step program. **Column 3** identifies the core principle or virtue that each Step emphasizes [15,33]. There are additional principles associated with each Step but many use this as a simplified summary. **Column 4** indicates the correlation coefficient between the ratio of length of abstinence/time in OA (**A**) or relapse frequency (**R**) and progress on/in- or increased time spent devoted to the step (*p* < 0.01), as identified by a 2002 PhD dissertation survey completed by 229 active OA members who self-identify as binge eaters (n = 194) or bulimics (n = 35) in a U.S. metropolitan area [27]. **Column 5** identifies the timeline by which each Step is typically performed or occurs. Steps 1–3 are often immediate and occur within the first 7–90 days of recovery. Steps 4–9 are often undertaken during the first year of recovery and revisited regularly throughout the recovery process. Steps 10–12 are undertaken daily as a foundation for a new way of life.

**Table 2 ijerph-18-07303-t002:** The Nine Tools of Overeaters Anonymous.

Tool Number and Description [20,21,22,34]		Frequency of Use [20,21,22,34]	Abstinence/Relapse Correlation [27]
Tool 1	**Plan of eating:** Adherence to a meal plan provided by a healthcare provider and discussed with a sponsor so as to eliminate problematic or “addictive/alcoholic” foods that cause craving and lead to overeating [20,21,22,34].	Most OA members believe this tool must be used daily and consistently, just as an alcoholic must consistently abstain from alcohol.	Adherence to food plan: A: 0.29 **, R: 0.34 ** Weigh/measuring food: A: 0.21 **, R: 0.23 **
Tool 2	**Sponsorship:** Working with a sponsor to reinforce OA tool use while progressing through OA’s Twelve Steps [20,21,22,34].	Can vary from daily at a set time to weekly or sporadically and unscheduled.	A: 0.14 *, R: 0.2 **
Tool 3	**Telephone calls** to other OA members to build social support and “talk program” [20,21,22,34].		A: 0.25 **, R: 0.29 **
Tool 4	**Anonymity:** This tool and Tradition (Tradition 12) protects participant confidentiality, enabling members to share honestly and openly without fearing gossip [15,18,20,21,22,34]. Promotes focus on one’s own spiritual and emotional growth (vs. external circumstances), and placing principles above personalities [15,18,20,21,22,34].	Ongoing, though some individuals and groups are more mindful of anonymity than others.	N/A
Tool 5	**Attending meetings:** Offers opportunities for fellowship, service, and a sense of connection to something greater than one’s self [20,21,22,34].	May vary from daily (especially within the first 90 days) to weekly or sporadically.	A: 0.14 *, R: NS
Tool 6	**Reading twelve-step literature:** Promotes growth and recovery. Provides guidance on the twelve-step process [20,21,22,34,35,36,37,38].	Can vary from daily to sporadically or as needed to work through OA’s 12 steps and to manage thoughts and emotions [20,21,22,34,35,36,37,38].	A: NS, R: 0.2 **
Tool 7	**Reflecting and writing** on one’s daily reading, thoughts, feelings, and twelve-step work. Writing is often read (“given away”) to one’s sponsor for feedback [20,21,22,34,35,36,37,38].		A: 0.16 *, R: 0.29 **
Tool 8	**Service**: Promotes selflessness, ego-reduction, and altruism. May be performed at the individual, meeting/group, or national/international levels. OA believes there is no service too small, and one’s greatest service is his or her own abstinence [20,21,22,34,39,40].	Frequency of formal service can vary from daily (abstinence) to weekly or monthly for meeting/national service positions or as a remedy for self-obsession.	A: 0.26 **, R: 0.3 *
Tool 9	**Action plan**: Some meeting formats provide specific instructions on daily use of each tool as part of one’s abstinence (and action plan). An action play may also include planning time for meal preparation, exercise, meeting with healthcare professionals, and prayer and meditation (part of Step 11) [20,21,22,34].	Can vary from daily or weekly to sporadically or as needed. Some members and groups may not use a formal action plan.	N/A

**Columns 1 and 2** identify each of the Nine Tools suggested by Overeaters Anonymous. **Column 3** provides a brief description of the frequency with which each tool is used. **Column 4** indicates the correlation coefficient between the ratio of length of abstinence/time in OA (**A**) or relapse frequency (**R**) and Tool use (* *p* < 0.05, ** *p* < 0.01, NS = not significant), as identified by a 2002 PhD dissertation survey completed by 229 active OA members who self-identify as binge eaters (n = 194) or bulimics (n = 35) in a U.S. metropolis [27].

**Table 3 ijerph-18-07303-t003:** The Twelve Traditions of Overeaters Anonymous.

The Twelve Traditions of Overeaters Anonymous	
Tradition 1	Our common welfare should come first; personal recovery depends upon OA unity [15].
Tradition 2	For our group purpose, there is but one ultimate authority—a loving God as He may express Himself in our group conscience. Our leaders are but trusted servants; they do not govern [15].
Tradition 3	The only requirement for OA membership is a desire to stop eating compulsively [15].
Tradition 4	Each group should be autonomous except in matters affecting other groups or OA as a whole [15].
Tradition 5	Each group has but one primary purpose—to carry its message to the compulsive overeater who still suffers [15].
Tradition 6	An OA group ought never endorse, finance, or lend the OA name to any related facility or outside enterprise, lest problems of money, property, and prestige divert us from our primary purpose [15].
Tradition 7	Every OA group ought to be fully self-supporting, declining outside contributions [15].
Tradition 8	Overeaters Anonymous should remain forever non-professional, but our service centers may employ special workers [15].
Tradition 9	OA as such, ought never be organized; but we may create service boards or committees directly responsible to those they serve [15].
Tradition 10	Overeaters Anonymous has no opinion on outside issues; hence the OA name ought never be drawn into public controversy [15].
Tradition 11	Our public relations policy is based on attraction rather than promotion; we need always maintain personal anonymity at the level of press, radio, films, television, and other public media of communication [15].
Tradition 12	Anonymity is the spiritual foundation of all these traditions, ever reminding us to place principles before personalities [15].

The Twelve Traditions of Overeaters Anonymous (OA) [15] are based on AA’s Twelve Traditions [18], and provide guidelines for OA governance and fellowship.

Altogether, OA’s Nine Tools, Twelve Steps, and Twelve Traditions can facilitate structure (e.g., use of the meal- and action planning tools), provide social support (e.g., use of sponsorship, telephone calls, and meeting attendance), enable open, anonymous sharing (e.g., through anonymity, sponsorship, telephone calls, reading/writing, and meeting attendance) to foster a sense of connection and belonging, and promote healing and recovery [20,21,22,27,34,41,42,43,44]. However, high-quality research on OA is required to better understand what aspects of OA are beneficial in curbing binge eating behaviors (if any), and what aspects of the program its members find helpful (and why).

## 2. Demystifying Overeaters Anonymous

### 2.1. Is Overeaters Anonymous a Religious Program? “God,” “Higher Power”, and Spirituality in OA

Capitalization and use of the word “God” and the pronouns “He/Him/His” prescribed to “God” appear throughout AA and OA literature [14,15,19,45]. Furthermore, OA’s second Tradition (Table 3) offers an explicit description of “one ultimate authority—a loving God… [15]” that is in line with religious doctrine and reveals AA’s Christian Evangelical beginnings [19,46,47]. Appendix A describes AA’s religious beginnings and provides excerpts from AA’s basic text on its viewpoints regarding spirituality and spiritual experiences, which extend also to OA. Although OA’s twelve steps have spiritual components, OA is not religious [15]. The OA preamble states “OA is not affiliated with any public or private organization, political movement, ideology, or religious doctrine [and takes] no position on outside issues” [15]. Some members find connection or reconnection with pre-established religious or non-religious spiritual beliefs during their recovery process [19]. However, many members successfully work the program while maintaining atheist or agnostic inclinations [15]. AA’s basic text shares the story of how AA’s founder was invited to “choose [his] own conception of God [47]” and OA literature shares this invitation with its readers [14,15]. Nevertheless, the intermingling of religious beginnings with a fundamentally spiritual approach may provide a barrier to entry and therapeutic success for some. Perhaps for this reason, some believe that spiritual programs work best when they are entirely disentangled from any for-profit motives or private entities. However, there is a rich history of integration between the two.

### 2.2. How Is Overeaters Anonymous Used for Treatment? Variety in OA Implementation

The way in which OA is used as a treatment intervention can vary. OA can be used as an adjunct/component of a multi-disciplinary treatment program [23,24,25], as occurs in the “Twelve-Step Minnesota Treatment Model [48]” [23,24]. OA can also be sought out independently by an individual or referred as an adjunct or standalone treatment [14,16,26,27].

The “Twelve-Step Minnesota Treatment Model” was pioneered in the late 1940s by several addiction treatment centers in Minnesota (e.g., Hazelden) [48]. This model employs 3–8 weeks of residential treatment with: interdisciplinary treatment staff; twelve-step program implementation in which patients work through the first 3–5 steps during admission (Table 1); a holistic, multiphasic disease concept and recovery approach; a drug-free treatment environment; aftercare involving twelve-step group/meeting attendance [48,49]. The efficacy of this model has been demonstrated in a variety of studies for substance-related and addiction disorders [50,51,52] and eating disorders [23,24,53].

### 2.3. Does Everyone Work the Same Program in the Same Way? Heterogeneity within OA

Most OA groups, formats, and sponsors suggest that members: (1) abstain from problematic foods and/or eating behaviors; (2) work through OA’s Twelve Steps with a sponsor (Table 1); (3) use OA’s Nine Program Tools (Table 2) to reinforce their recovery [13,14,15,16,17,22,27,30]. However, variety exists in the formatting, structure/framework, and degree to which the program is used at the group, meeting, and individual levels [27,30,54]. For example, sponsors and sponsorship vary significantly within OA, contributing to variation in individual OA recovery.

Such heterogeneity creates challenges when collecting and compiling data on OA, as interventions are non-uniform. For example, some groups provide operational definitions of “abstinence/sobriety” that its group members all agree to follow [30,54]. These definitions may include abstaining from certain foods (e.g., white sugar, flour, alcohol) and/or food behaviors (e.g., compulsive eating, overeating, undereating/restricting, and/or binge/purge behaviors), and/or weighing, measuring, and pre-committing all food [30,54]. Some groups suggest an individual define his/her own abstinence with or without the help of a sponsor, nutritionist, and/or healthcare professional [14,30,54]. Some groups and sponsors provide specific instructions or suggestions on how each of OA’s Twelve Steps and Nine Tools should be used, while others do not [17,20,27,30,54]. Therefore, some OA members use OA’s Tools, Steps, and Traditions daily, whereas others use them sporadically (Table 1, Table 2 and Table 3) [27]. Some groups provide specific reading and writing activities that guide individuals through the Twelve-Step literature and Steps; others do not [17,22,30,35,36,37,38]. Some groups and meetings focus entirely on specific OA or AA literature; some meetings provide speakers or focus on specific topics/populations such as newcomers or relapse [30,54,55]. Some groups provide abstinence and time requirements for anyone who wishes to share at meetings; others are open to discussion from all. This variability can contributes to misunderstandings and over-generalizations regarding OA’s efficacy, what OA is, and how it can be used [41].

### 2.4. Is Overeaters Anonymous Synonymous with the Grey Sheet Diet? Variety in Twelve-Step Eating Disorder Programs

Outside of OA, a variety of different twelve-step programs exist for eating disorders that are sometimes confused with OA. It is important for researchers, clinicians, and patients to understand which programs and program elements *are-* and *are not* associated with OA. Many clinical criticisms of OA pertain to programs or program aspects that are not actually OA-endorsed. Furthermore, some patients may be better suited for twelve-step eating disorder recovery programs that exist outside of OA. 

To highlight the risk of confusing other programs with OA, we provide the following example: another twelve-step fellowship of compulsive overeaters, separate from OA, endorses use of a particular meal plan (the “grey sheet” meal plan) that was recommended by OA between 1960–1980 [56]. The meal plan received criticism for endorsing carbohydrate restriction and demonization, creating unrealistic “food rules” that are associated with binge eating (see Section 4.2), and opposing clinical beliefs that nutrition and meal planning should be unique to each individual (see Section 4.2). As a result of these shortcomings, the OA World Service Business Conference decreed in 1997 that “offering food plans at OA meetings is a violation of Tradition 10… [and]… OA as a whole cannot print, endorse, or distribute food plan information to its members… [but] ought best concern [itself] with [its] suggested program of recovery—the Twelve Steps (*Rescinded 2000*) [57],”(see Table 3 for Tradition 10) [56]. 

Additional twelve-step recovery groups for eating disorders that exist outside of OA appear to have different food philosophies and cultures. Examples include: Anorexics and Bulimics Anonymous (ABA); Compulsive Eaters Anonymous (CEA); CEA-Honesty, Openness, Willingness (CEA-HOW); Eating Disorders Anonymous (EDA); Food Addicts Anonymous (FAA). The importance of getting patients in touch with the right program and fellowship for them is described in Section 4. Overall, it is important to understand that: (1) other twelve-step recovery programs for eating disorders exist that should not be considered synonymous with OA; (2) OA does not endorse any particular food plan; (3) offering food plans at OA meetings is considered a violation of OA’s 10th Tradition [57].

## 3. Research on Twelve-Step Interventions

Binge eating disorder—in the same way as substance-related and addictive disorders—has been described as a chronic relapsing disorder characterized by preoccupation and craving, impaired control of consumption, and social impairment, among other similar features [2,58,59,60,61]. These commonalities can be particularly helpful in understanding why a twelve-step intervention traditionally used for substance-related and addictive disorders would be considered for use in treating binge eating disorder. Tools for assessing food addiction subtypes have recently been described [61,62,63,64,65], and include the Yale Food Addiction Scale v2.0 (YFAS v2.0) [61,64,65] and the Disordered Eating Food Addiction Nutrition Guide (DEFANG) [62,63] (see Section 4.2).

### 3.1. Research on Twelve-Step Interventions

Numerous studies show that twelve-step interventions are effective in promoting short- and long-term resilience and recovery from substance-related and addictive disorders, both when used as a component of a larger treatment model and/or when sought out independently as a standalone treatment [50,51,52,66,67,68,69,70,71]. For this reason, the National Institute on Drug Abuse (NIDA)’s 2018 research-based guide to drug addiction treatment reports that most drug addiction treatment programs encourage participation in twelve-step programs after formal treatment because twelve-step programs offer an additional layer of community-level social support that helps patients “achieve and maintain abstinence and other healthy lifestyle behaviors over the course of a lifetime [66].” 

In 1997, the National Institute on Alcohol Abuse and Alcoholism (NIAAA) sponsored two parallel but independent, randomized clinical trials that randomly assigned adults receiving outpatient therapy (*n* = 952) or aftercare (*n* = 774) for alcohol dependence to receive 3 months of either twelve-step facilitation (TSF, outpatient *n* = 312, aftercare *n* = 226), cognitive behavioral therapy (CBT, *n* = 283 and 244, respectively), or motivational enhancement therapy (MET, *n* = 284 and 240, respectively)(Project MATCH) [72]. Participants were followed for up to 15 months. In the outpatient group, 35.6% of TSF participants reported “no drinking” at the 15-month follow-up period, which was significantly higher than the corresponding percentages for CBT (24.7%) and MET (30.3%) (Bonferroni-corrected *p* = 0.0024). Outpatient TSF clients also had significantly higher amounts of time before taking their first drink, with 24% of individuals receiving TSF avoiding any drinking in months 4–15, while corresponding percentages for CBT and MET were 15% and 14%, respectively (proportional hazards analysis *p* = 0.0001). In the aftercare group, TSF produced slightly higher increases in the percentage of days abstinent and in reduced drinking days relative to CBT or MET “toward the end” of the 15-month follow-up period. For example, at the 9-month follow-up, TSF produced 87% of days abstinent, whereas CBT produced 73% (linear *p* < 0.001) [72]. These and other findings suggest that twelve-step participation can help promote recovery from substance-related and addictive disorders, and may have potential for individuals with binge eating disorder, particularly those who view their disorder as an addiction with a strong component of impaired control over consumption (loss of control).

### 3.2. Research on Overeaters Anonymous

Research on the effectiveness of OA is limited to lower levels of evidence that include gray literature (a 2002 dissertation [27]), case reports [24,63], case series [23], and some qualitative studies [41,73,74]. These lower levels of evidence are generally supportive of OA’s effectiveness, and are building toward the need for higher quality research (e.g., randomized controlled trials). The available data on OA’s effectiveness as a complementary or integrated intervention for weight and eating disorders will be presented here. 

OA’s first tool, “Plan of Eating”, (Table 2) [17,20,21,22] is similar to the “regular eating” intervention introduced as a primary aim of Stage One in cognitive behavioral therapy [75]. This tool is considered to be “fundamental to successful treatment [75]” and “the foundation upon which other changes are built [75]” in cognitive behavioral therapy and OA alike [17,20,21,22,27,75]. Within OA, a 2002 PhD dissertation study identified adherence to a food plan as one of three behaviors most consistently associated with abstinence from compulsive eating in the existing literature base (the other two behaviors were attending a support group and long-term individual psychotherapy) (Bonato & Boland, 1987 [76]; Kayman et al., 1990 [77]; Perri et al., 1988 [78], as cited in Kriz, 2002) [27]. 

The same 2002 dissertation study conducted a survey among 229 active OA members who self-identified as binge eaters (*n* = 194) or bulimics (*n* = 35) in a U.S. metropolitan area [27]. The study found frequency of use of the tool “Plan of Eating” to correlate positively with the ratio of length of abstinence/time in OA (Pearson product-moment correlation coefficient (*r*) = 0.29, *p* < 0.001) and inversely with relapse frequency (*r* = 0.34, *p* < 0.01) (Table 2) [27]. Frequent use of OA’s other Steps and Tools were also found to correlate positively with abstinence ratio and inversely with relapse frequency (see Table 1 and Table 2 for the correlation coefficient and significance levels associated with each of the individual Steps and Tools) [27]. These findings will need to be verified by higher quality, peer-reviewed clinical trials.

Research also supports OA’s effectiveness when integrated into a standard treatment model. For example, a private Israeli treatment center using a twelve-step program “as an adjunct to counseling and treatment” over an average 7-month period in adult women with obesity and bulimia nervosa (*n* = 578) between 1994–1999 observed a mean weight loss of 9.7 kg among adults with obesity (*n* = 409) and a 71% success rate in cessation of purging behaviors among adults with bulimia (*n* = 169) over a 6-month period [23].

Two case studies also exist [24,63]. A residential treatment center in the U.S. reported a 397-pound female whose treatment incorporated OA [63]. The patient reported previously using compulsive eaters anonymous honesty, openness, willingness (CEA-HOW, see Section 2.3) for 6 months to achieve a 150-pound weight loss and freedom from food addiction and its associated mental obsession with food and eating behaviors, but lost both outcomes when she moved and stopped working the program. She reported being “[unable to] put 30 days [of recovery] together” independently. Her residential and outpatient treatment included: a balanced meal plan (three meals plus two–three snacks from all food groups) that avoided addictive foods; exercise; psychotherapy; involvement with an OA group and sponsor. She stopped bingeing, lost weight (30 lbs. during her 60-day inpatient treatment and another 50 lbs. as an outpatient), and maintained her personal definition of recovery, but relapsed when her therapist suggested she eat “all foods in moderation.” At her 2-yr follow-up, she re-engaged with CEA-HOW and was optimistic.

A private addiction treatment center in Spain also used the “Minnesota Model” (addressed in Section 2.2) to treat a patient with binge eating/purging anorexia nervosa [24]. The patient’s twelve-step “sobriety” was defined as eating all five planned daily meals/snacks with nothing excluded. She worked through OA’s Twelve Steps with a sponsor during inpatient and outpatient care. During her 3-month inpatient treatment, she gained 1.3 kg (3 lbs.), achieving a 0.46 kg/m^2^ increase in BMI (weight: 46.2kg/101.8 lbs.; BMI: 16.21 kg/m^2^ at discharge). During her subsequent 8-month outpatient treatment, she regained normal menstruation and gained an additional 8.1 kg (17.9 lbs.) and 2.28 kg/m^2^ increase in BMI to achieve weight restoration at 54.3 kg (119.7 lbs.; BMI: 19.05 kg/m^2^). She had one relapse 5 months into her outpatient treatment when she was traveling and did not have telephone service to access her twelve-step fellowship, reinforcing the effectiveness of the twelve-step intervention.

Qualitative studies also report subjective experiences of receiving a sense of belonging [41] as well as transformations from a “dieting mentality” to “emotional and spiritual recovery” [73] with freedom from weight concerns and compensatory behaviors [74].

These published accounts of OA’s effectiveness emphasize the ability of OA—in the same way as other twelve-step interventions—to be used as an adjunct or component of a multi-disciplinary treatment program [23,24,25,48] and/or as an independent treatment referred or sought out by the individual [26,27,54]. However, randomized controlled trials are needed in order to empirically determine OA’s effectiveness for clinical use. It will be important for such trials to be conducted in a way that protects OA’s tradition of anonymity (Tool 4, Table 2; Tradition 12, Table 3), which enables members to share honestly and openly without fearing gossip, thus deepening social connections and providing potentially therapeutic benefits [15,18,20,21,22,34]. Although OA’s tradition of anonymity may pose a challenge for empirical research, it is one that research on other twelve-step interventions have overcome [72].

## 4. Clinical Criticisms Against Overeaters Anonymous

### 4.1. Self-Selection and Internal Biases: The Importance of Proper Classification and Use

Evidence reviewed herein suggest that OA may be an underutilized tool in the treatment and recovery of some forms of binge eating. Clinical experience suggests that for properly classified individuals (e.g., those with food addiction in the absence of pathological dietary restraint [62,79,80]) this option may be a wise and cost-effective choice for sustaining long-term recovery. Meanwhile, outcomes are likely to depend on whether individuals find a twelve-step group appropriate for their particular condition. Importantly, the individual seeking help may not always be in the position to make this assessment. For example, high levels of internalized weight bias [81] may in turn lead to the prioritization of weight loss over other recovery-related matters. Many individuals seeking support through OA may be attempting to manage a perceived crisis of “living in a body that does not feel like home.” This may be linked to forms of early life adversity and/or post-traumatic stress disorder [62,79,82] that can skew one’s perception of the primary presenting issue.

Consequently, misclassification issues can arise through self-selection processes. The emphasis on “self-diagnosis” can lead individuals to identify their problem and (consciously or subconsciously) seek out solutions that match goals related to weight loss and/or ego-syntonic eating disorder pathology (e.g., the relentless pursuit of thinness). This can be true of any treatment intervention one pursues, clinical or non-clinical [83,84]. In some cases, this can exacerbate eating disorder symptoms in the long run [85]. Many registered dietitian nutritionists who work in treatment settings have reported this phenomenon (anecdote per DAW) [86]. To illustrate further, many restrictive eaters tend to view bingeing as their target behavior change and are often unable to recognize the contribution of dietary restraint to their binge eating [80]. Such individuals can score high on measures of body dissatisfaction, even at BMIs near or below average [62]. These individuals may seek out sponsors/fellows without a similar history of restrained eating who endorse cutting out entire food groups, reducing calories for the sake of weight reduction, and using behavioral strategies to eat less food. Clinicians in the field have seen such inadvertent influences develop into full-blown anorexia nervosa and bulimia nervosa (anecdote per DAW) [86]. Therefore, a format of OA that incorporates abstinence from both undereating and overeating, regardless of an individual’s weight and/or diagnosis, may be best suited for some individuals, at least initially (see the OA Honesty, Openness, Willingness (OA-HOW) format identified in Appendix A as an example).

### 4.2. Concerns Related to Nutritional Guidance

Many conventional eating disorder treatment professionals tend to dissuade patients from use of OA because of concerns related to nutritional guidance. Specifically, members may give food advice that is generic (or based on what has worked for them) and leads some members astray, despite good intentions. One consensus is that the culture of many twelve-step programs is overly restrictive, contains punitive food messaging, and may even be harmful to one’s long-term “relationship to food”. Conventional eating disorder approaches tend to encourage eating all foods in moderation (“inclusion”) rather than reducing exposure to foods that produce “uncontrollable craving”, such as highly palatable foods (“exclusion”). This model assumes that dietary restraint is the primary driver of binge eating [80,87], whereas addiction models assume that addictive foods or food behaviors are the primary driver of binge eating, with dieting behavior being a consequence rather than a cause [62,63,80]. There appears to be an important and timely need to find areas of intersection between these theoretical frameworks. Recently, efforts have been made to help clinicians discern broadly which nutrition approach may be most appropriate, using a combination of assessment tools, clinical intuition, contextual factors, and collaboration with the patient and multi-disciplinary treatment team [62,63]. The Yale Food Addiction Scale v2.0 [61,64,65] and the Disordered Eating Food Addiction Nutrition Guide (DEFANG) [62,63] are two examples of clinical tools that can be used to assess eating disorder subtype and screen for food or eating addiction.

### 4.3. Overlooked Effects of Weight Stigma and “Fatphobia”

A final criticism of OA is the potential for perpetuation of weight stigma, which has emerged as an important component of obesity context [79,88]. Weight stigma can generate societal pressures to pursue thinness and contribute to poor body image, weight suppression, and full-blown eating disorders [89,90]. Described as a vicious cycle wherein weight stigma begets weight gain [91], feedback loops include stress and maladaptive eating behaviors [92]. Some authors have suggested that weight stigma is an overlooked cause of the obesity epidemic [93]. Surprisingly to some, efforts to suppress weight can have the opposite effect [94,95]. Weight stigma and the rationale for weight-inclusive health policy has become an increasingly important topic in public health discussions surrounding obesity [88]. Given this position, interventions may be more rightly aimed at altering the behaviors and attitudes of those who stigmatize, rather than toward the stigmatized individuals.

Many eating disorder professionals have identified twelve-step groups as a source of weight stigma and “fatphobia”, but there is likely a selection bias occurring: professionals are likely to hear from individuals reporting adverse outcomes with OA and not from the individuals who succeeded with it. Some OA formats place limits and structure around the frequency with which its members weigh themselves (e.g., 1/month until goal weight is achieved and 1/week on maintenance, or as directed by a healthcare professional, in order to provide this information to health professionals) [42,43,44]. This structure can either serve to desensitize the individual to his/her weight or can exacerbate weight hypersensitivities. Research is needed to investigate associations between OA and weight stigma from the perspectives of OA members, the lay public, and from treatment professionals.

### 4.4. Lack of High-Quality Empirical Evidence

As addressed in Section 3.2 (“Research on Overeaters Anonymous”), research on the effectiveness of OA is limited to lower levels of evidence that include gray literature (a 2002 dissertation [27]), qualitative studies [41,73,74], case reports [24,63], and case series [23]. These lower levels of evidence are supportive and are building toward the need for higher quality research (e.g., randomized controlled trials). Section 1 and Section 2 emphasize that OA is a multi-faceted program. There are many aspects of OA that may be beneficial in curbing binge eating behaviors. Table 2 and Table 3 highlight some initial work that has been done to identify which aspects of the OA program may be beneficial to its members [27]. However, this initial work lacks peer review or proper testing. Therefore, it will be important for high-quality research on OA to be conducted, including randomized controlled trials to empirically determine OA’s effectiveness for clinical use. It will also be important for OA’s theoretical/scientific basis to be better understood so that its hypotheses can be formulated and tested. Furthermore, high-quality research on OA will also be required to better understand what aspects of OA are beneficial for reducing binge eating behaviors, what aspects of the program its members find helpful (and why), and who the program works well for (and why). It will be important for such studies to be conducted in a way that protects OA’s tradition of anonymity (Tool 4, Table 2; Tradition 12, Table 3), as has been the case in research studies on other twelve-step interventions [72].

### 4.5. Critical Clinical Conclusions

It is important to acknowledge that all twelve-step programs have limitations and have been subject to similar criticism. The question is whether the benefits outweigh the potential risks. Our primary argument is that the answer to this question will depend on proper assessment and classification of the individual, as well as the type of OA intervention engaged, and what other treatment services are being received. More research is needed to identify which eating disorder phenotypes and subtypes are best suited for OA, and if better discernment with the help of clinicians can lead to better health outcomes. It may also prove valuable to identify outcomes other than weight and abstinence time, such as various mental health quality of life measures [96,97].

## 5. Conclusions

Overeaters Anonymous provides a promising complement for treating binge eating disorder. OA is non-professional (Tradition 8) and is not affiliated with any outside enterprise (Tradition 6, Table 3) [15]. However, OA can be referred clinically or sought out by an individual confidentially, without a diagnosis, and is free of charge, thus overcoming the major barriers of stigma, misconception, lack of diagnosis, and inadequate insurance coverage that prevent many individuals with binge eating disorder from receiving treatment [4,6]. OA’s Nine Tools, Twelve Steps, and Twelve Traditions can help provide structure and social support [20,21,22,27,34,41,42,43,44], which individuals with eating disorders have especially valued during the COVID-19 pandemic [10]. Furthermore, OA provides >955 online and telephone meetings (in addition to >3500 face-to-face meetings) in multiple languages (including ESL) [28,29], making it a viable therapeutic intervention throughout the COVID-19 pandemic, and likely appealing to those who appreciate the convenience and accessibility of tele-health treatment modalities [12]. Further clinical research on the efficacy of this intervention is urgently needed for practitioners to better understand who can benefit most from OA and how OA can be integrated into clinical practice. Randomized controlled trials are also needed to empirically determine OA’s effectiveness for clinical use. It will be important for future research to be conducted in a way that protects OA’s tradition of anonymity and the therapeutic benefits it affords [15,18,20,21,22,34], as has been achieved in research on other twelve-step interventions [50,51,52,67,68,70,71,72].

## Data Availability

The data presented in this study are openly available in Zenodo (zenodo.org, accessed on 4 November 2020) at doi:10.5281/zenodo.5075531, reference number 5075531. Available online: zenodo.org/deposit/5075531#, accessed on 4 November 2020.

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
