# Peer review of "Overeaters Anonymous: An Overlooked Intervention for Binge Eating Disorder"

_ijerph, 2021, doi:10.3390/ijerph18147303_

Round 1

Reviewer 1 Report

This Reviewer read the revised version of this manuscript with interest.

Abstract – According to the authors OA is a 12 step programme for treating BED and BED is similar to substance-related and addictive disorders ”processes at the genetic, neurobiological and behavioral levels”.

Please describe the 12 steps of OA, the ”processes” that are similar and the data that supports the hypothesis that OA improves outcomes.

  1. Introduction – This is a clearly written without using too many words, a good thing.
  2. Binge Eating … - While the text has been abbreviated, another good thing, the fact remains that nothing new is presented. The authors may be right in that some readers might profit from a brief update on overlapping brain mechanisms for BED-addiction. But the text is not very helpful in merely listing brain areas/neurotransmitters that undoubtedly play a role in eating and addiction, but also in all other kinds of behaviour. In reviewing an area of research, authors should come up with an interesting new interpretation, rather than, as these authors do, merely listing previous areas of research. Thus, the insert below line 140 adds confusion rather than clarity by listing at least 12 signalling molecules without saying what these might do, just that they are ”involved”.
  3. Overeaters Anonymous: …- This paragraph jumps from the brain to describing the 12 steps of OA, a big jump into the world of the "Powers of God", "ego-reduction" and "altruism"! This is described in an insert starting after line 140 and continuing through line 155. This Reviewer does not understand what the ”Efficacy” column might mean. There should be data involved and these should be brought out into daylight.

The authors may be right in that OA is helpful in treating BED. But it remains entirely unclear how the previous paragraph on the brain and related biology can inform readers that this may be so.

  1. Are Twelve-Step …- This paragraph is supposed to offer support for the possibility that OA might work but the text is scrambled with ”corrective emotional experiences”, ”interpersonal relationships”, ”childhood rejection”, ”spiritual experience” and more similar stuff that cannot (as yet, if ever?) be related to the little we know about the brain. Because of this lack of connection, this Reviewer was not surprisied to read that the APA has recognized a role of ”sprituality” and ”religious issues” in some of their mental disorders. But these are non-scientific speculations all too common in pre-scientific times.

Back to neurobiology the authors call attention to an overview of the activity of the brain associated with meditation and prayer and this is thought to be important in OA. To strengthen the case, the authors refer to a paper on bulimics ingesting a chocolate milk shake. This discussion transgresses the bounds of scientific analysis, conceptual analysis and even common sense.

No connection has been convincingly demonstrated between science and OA and the authors´ analysis goes way beyond what can be said at the present state of knowledge.  

Author Response

24 June 2021

Dear Reviewers –

We thank you for your thoughtful review and for your enthusiasm. We have attempted to address all of your comments and believe that your comments make this manuscript stronger and more relevant.

We have itemized the comments and our attempts to address them below. Please note that current responses are addressed in BLUE and previous responses are included in GREEN. Each reviewer’s responses are provided together, with the reviewer identified in DARK BLUE (ex: “Reviewer 1,” “Reviewer 2,” etc.).

Note that we have also added two sentences at the end of the introduction that bring current relevance to the field. These were not sentences suggested by the reviewers. However, binge-eating disorder is relevant to COVID lockdown. Thus, we felt this addition strengthened the manuscript.  

*

Reviewer 1

Please describe the 12 steps of OA, the ”processes” that are similar and the data that supports the hypothesis that OA improves outcomes.

The 12 steps of OA are included in Table 1. To our knowledge, OA has not been studied in a randomized controlled trial. The data that is available is located in the manuscript in section 4.1, beginning on line 459.

  • Introduction – This is a clearly written without using too many words, a good thing.

  1. Thank you for this comment.

  • Binge Eating … - While the text has been abbreviated, another good thing, the fact remains that nothing new is presented. The authors may be right in that some readers might profit from a brief update on overlapping brain mechanisms for BED-addiction. But the text is not very helpful in merely listing brain areas/neurotransmitters that undoubtedly play a role in eating and addiction, but also in all other kinds of behaviour. In reviewing an area of research, authors should come up with an interesting new interpretation, rather than, as these authors do, merely listing previous areas of research. Thus, the insert below line 140 adds confusion rather than clarity by listing at least 12 signalling molecules without saying what these might do, just that they are ”involved”.

  1. Thank you for this feedback. We have removed the section on neurobiology of binge eating disorder and food addiction as it did not add anything new.

  • Overeaters Anonymous: …- This paragraph jumps from the brain to describing the 12 steps of OA, a big jump into the world of the "Powers of God", "ego-reduction" and "altruism"! This is described in an insert starting after line 140 and continuing through line 155. This Reviewer does not understand what the ”Efficacy” column might mean. There should be data involved and these should be brought out into daylight.

  1. Thank you. We have removed the “neurobiology of BED” section, so the intro is followed by information on OA.
  2. The efficacy column in Tables 1 and 2 have been changed to “Abstinence/Relapse Correlation[1],” and the description in the Table legends have been elaborated: “Column 4 indicates the correlation coefficient between the ratio of length of abstinence/time in OA (A) or relapse frequency (R) and progress on/in the step (for Steps 4 and 9) or increased prayer and meditation (for Step 11)(p < 0.01), as identified by a 2002 PhD dissertation survey completed by 231 active OA members in a metropolitan area in the U.S.[1]. .”

  • The authors may be right in that OA is helpful in treating BED. But it remains entirely unclear how the previous paragraph on the brain and related biology can inform readers that this may be so.

  1. We thank the reviewer for this insight. The purpose of that section had been to help an uninformed reader understand why a 12-step treatment, traditionally used for substance-related and addiction disorders, would be considered for binge eating (by identifying neurobiological similarities that exist in SRADs and BED). However, we have removed that section to help simplify and streamline the manuscript as a whole.

  • Are Twelve-Step …- This paragraph is supposed to offer support for the possibility that OA might work but the text is scrambled with ”corrective emotional experiences”, ”interpersonal relationships”, ”childhood rejection”, ”spiritual experience” and more similar stuff that cannot (as yet, if ever?) be related to the little we know about the brain. Because of this lack of connection, this Reviewer was not surprisied to read that the APA has recognized a role of ”sprituality” and ”religious issues” in some of their mental disorders. But these are non-scientific speculations all too common in pre-scientific times.

  1. Thanks for this insight. We have removed much of the content you mentioned (re: corrective emotional experiences, etc), so that paragraph now reads simply: “Qualitative studies also report subjective experiences of receiving a sense of belonging[2] as well as transformations from a “dieting mentality” to “emotional and spiritual recovery”[3] with freedom from weight concerns and compensatory behaviors[4].”

  • Back to neurobiology the authors call attention to an overview of the activity of the brain associated with meditation and prayer and this is thought to be important in OA. To strengthen the case, the authors refer to a paper on bulimics ingesting a chocolate milk shake. This discussion transgresses the bounds of scientific analysis, conceptual analysis and even common sense.

No connection has been convincingly demonstrated between science and OA and the authors´ analysis goes way beyond what can be said at the present state of knowledge.  

  1. Thank you for this feedback. Per your comments, this section has been removed.

Reviewer 2 Report

Old Comments:

Overeaters Anonymous: An Overlooked Intervention for Binge 2 Eating Disorder

Line 26 – “1-5% of the general population[2,3]” this is based on two references: [2] Kessler et al., 2013 and [3] Hutson et al. 2018

Kessler et al. state: “lifetime prevalence estimates average 1.0% for BN and 1.9% for BED across surveys” and “Country-specific lifetime prevalence estimates are consistently (median; interquartile range) higher for BED (1.4%; 0.8-1.9%)”

Hutson et al. state; “2-5% of the adult population and is more common in females than males (Dingemans et al., 2002, Kessler et al., 2013).” The Dingemans review was undertaken almost 20 years ago and at a time when BED is not a formal diagnosis within the then DSM-IV. Whilst estimating prevlance is difficult, it is important not to re-use figures from 2002 without explaining their limitations and to present the latest prevalence estimates clearly.

Authors response: We thank you for pointing this out. We have changed the statement to exclude the statistic from the Dingemans review. The statement now reads “with an estimated lifetime prevalence of 1.9% globally (Kessler et al) and 2.8% in the U.S. (Hudson et al). The “2.8% in the U.S.” is taken from Hudson et al.’s 2007 publication of “the prevalence and correlates of eating disorders in the National Comorbidity Survey replication.” In the survey and publication, the researchers analyzed data from the National Comorbidity Survey Replication (NCS-R). “NCS-R diagnoses were based on Version 3.0 of the World Health Organization Composite International Diagnostic Interview (CIDI) (Kessler and Ustun 2004), a fully structured lay administered diagnostic interview that generates diagnoses according to both ICD-10 and DSM-IV criteria” (Hudson et al., 2007). Notably, the CIDI criteria used for BED was in alignment with the current DSM-V diagnostic criteria for BED that requires the individual to have experienced at least 1 episode of binge eating weekly for at least 3 months of symptoms, and assesses for loss of control and marked distress through “a series of questions about attitudes and behaviors that are indicators of loss of control and of distress, rather than by direct questions,” (Hudson et al., 2007). Therefore, although the DSM-V diagnosis for binge eating disorder did not yet exist at the time of Hudson et al.’s 2007 publication, we feel that their diagnostic criteria was suitable to convey a relatively accurate measure of its prevalence at that time. 

New comment:

Unfortunately, this is still not clear enough.  From the manuscript PDF I can see, it does not read as you indicated but instead you have now written “Binge eating disorder affects ~2.8 million American adults…” and cited Hudson

  1. You should make it clear that these data are, firstly, not current but derive from data collected in 2001-2003 and, secondly, that even then they are estimates.
  2. The statistic is 2.8% (and you should indicate the US population at that time to make 2.8% meaningful)

Similarly, the Kessler estimates derive from data collected in 2008 and this should be made clear.

Old Comments:

Line 166: In this paragraph the concept “God” is explained but nonetheless it should be acknowledged that Tradition 2 of OA is quite explicit in its description “there is but one ultimate authority a loving God” and uses capitals for He etc. in line with religious doctrine.

Authors response: Thank you for pointing this out. We have acknowledged your point and have added some additional historical information on AA to set the context for its view on spirituality (1st and last paragraphs in section 5.1, as well as supplementary text in S1).

New comment:

In 5.1 the use of the word “grim” is not appropriate for a science article.

Whilst the text in the section “Spirituality in OA” provides details of the evolution and current stance on “God”, much of it feels like a defence of it rather than a critical review.  You acknowledge in the final line that the OA’s views are “incongruous” but in the context of it not being a barrier. This section should have fewer quotations (it is not a qualitative piece) and a critical analysis of the inconsistencies and this program, including this being a barrier to some.

Old Comments:

Line 239: “2.4. Variety in Twelve Step Programs for Eating Disorders”

I am not sure about the relevance of this section. It seems to describe whether meal plans are useful and gives the names of other programs. How do these relate to OA and how have they changed OA approaches/policy. This paper is about OA and descriptions of alternative programs and their history do not feel be useful as they are. I would recommend deleting unless it can be made relevant to OA. In any event the following should be clarified:

Line 243: provide a brief explanation of ““grey sheet” meal plan.”

Line 246: provide a brief description of “Dignity of Choice pamphlet”

Authors response: We thank you for expressing your receipt of this paragraph. The purpose of this paragraph was primarily to address common misconceptions that clinicians have of OA based on elements of other 12-step programs that are not affiliated with OA (ex: many dieticians incorrectly associate the “grey sheet diet” with OA, and therefore criticize OA for suggesting all members follow the same low-carbohydrate meal plan, when OA does not in fact endorse (but rather condones) the grey sheet diet. We have significantly restructured this section in order to make its purpose clearer; we have also provided brief explanations of the grey sheet meal plan and Dignity of Choice pamphlet (with citations to the actual contents of both). We hope this you will find these revisions satisfactory.

5.4. Is Overeaters Anonymous Synonymous with The Grey Sheet Diet? Variety in Twelve Step Eating Disorder Programs

Outside of OA, a variety of different twelve-step programs exist for eating disorders that are sometimes confused with OA. It is important for researchers, clinicians, and patients alike to understand which programs and program elements are- and are not associated with OA, as many clinical criticisms of OA pertain to programs or program aspects that are not actually OA-endorsed. For example, GreySheeters Anonymous (GSA) is a twelve-step fellowship of compulsive overeaters, separate from OA, who use the same “grey sheet” meal plan while working with a sponsor through OA’s twelves steps3. GSA and its namesake “gray sheet” diet/meal plan are often confused with OA as the “gray sheet meal plan” originated within OA in the late 1960s and was recommended by OA until it was replaced by the Dignity of Choice pamphlet4 in 19795. The “grey sheet meal plan” includes weight loss and maintenance options5,6; the meal plans’ restriction of carbohydrates opposes the U.S. Department of Agriculture’s recommended daily allowance of 130 g of carbohydrates per day with suggestions to “eat plenty of whole grains7.” In addition to risking nutrient deficiencies (ex: B vitamins, magnesium, and selenium that are found in whole grains)6, the carbohydrate restriction and overall restrictive nature of the meal plan6 have been criticized for risking carbohydrate demonization and creating unrealistic “food rules” that are associated with binge eating (see section 3.3). The Dignity of Choice book and pamphlet evolved from the grey sheet diet and offered 5 – 8 meal plan options to select from5,8. However, both the grey sheet diet and the Dignity of Choice meal plans oppose clinical beliefs that nutrition and meal planning must be unique to each individual (see section 3.3). As a result of these shortcomings, the OA World Service Business Conference decreed in 1997 that “offering food plans at OA meetings is a violation of Tradition 10… [and]… OA as a whole cannot print, endorse, or distribute food plan information to its members… [but] ought best concern [itself] with [its] suggested program of recovery – the Twelve Steps (Rescinded 2000)9,”(see Table 3 for Tradition 10)5.

Additional examples of twelve-step recovery groups for eating disorders that exist outside of OA include: Anorexics and Bulimics Anonymous (ABA); Compulsive Eaters Anonymous (CEA); CEA-Honesty, Openness, Willingness (CEA-HOW); Eating Disorders Anonymous (EDA); and Food Addicts Anonymous (FAA). All appear to have different food philosophy and culture. The importance of getting patients in touch with the right program and fellowship for them is described in section 3.3.

New comment:

This, again, reads like a defence of OA rather than a critical evaluation.  The point to me of this section in the context of a review on OA being an overlooked intervention, is to (more briefly) point out that OA may be overlooked as it is confused with other programmes.

Author Response

24 June 2021

Dear Reviewers –

We thank you for your thoughtful review and for your enthusiasm. We have attempted to address all of your comments and believe that your comments make this manuscript stronger and more relevant.

We have itemized the comments and our attempts to address them below. Please note that current responses are addressed in BLUE and previous responses are included in GREEN. Each reviewer’s responses are provided together, with the reviewer identified in DARK BLUE (ex: “Reviewer 1,” “Reviewer 2,” etc.).

Note that we have also added two sentences at the end of the introduction that bring current relevance to the field. These were not sentences suggested by the reviewers. However, binge-eating disorder is relevant to COVID lockdown. Thus, we felt this addition strengthened the manuscript.  

Reviewer 2 comments

  • New comment:

Unfortunately, this is still not clear enough.  From the manuscript PDF I can see, it does not read as you indicated but instead you have now written “Binge eating disorder affects ~2.8 million American adults…” and cited Hudson

  1. You should make it clear that these data are, firstly, not current but derive from data collected in 2001-2003 and, secondly, that even then they are estimates.
  2. The statistic is 2.8% (and you should indicate the US population at that time to make 2.8% meaningful)

Similarly, the Kessler estimates derive from data collected in 2008 and this should be made clear.

Thank you for pointing this out re: the old data. It’s clear that it’s time to update this data, However, with only this data available, we have chosen to remove it.

Old Comments:

Line 166: In this paragraph the concept “God” is explained but nonetheless it should be acknowledged that Tradition 2 of OA is quite explicit in its description “there is but one ultimate authority a loving God” and uses capitals for He etc. in line with religious doctrine.

Authors response: Thank you for pointing this out. We have acknowledged your point and have added some additional historical information on AA to set the context for its view on spirituality (1st and last paragraphs in section 5.1, as well as supplementary text in S1).

New comment:

In 5.1 the use of the word “grim” is not appropriate for a science article.

Whilst the text in the section “Spirituality in OA” provides details of the evolution and current stance on “God”, much of it feels like a defence of it rather than a critical review.  You acknowledge in the final line that the OA’s views are “incongruous” but in the context of it not being a barrier. This section should have fewer quotations (it is not a qualitative piece) and a critical analysis of the inconsistencies and this program, including this being a barrier to some.

Thank you for your comments. We have moved an abbreviated version on the historical context to supplementary material S1 (and replaced the word “grim” with “poor”). We have retained only objective information on religion and spirituality in OA and acknowledge that this could pose a barrier of entry to some.

  • Old Comments:

Line 239: “2.4. Variety in Twelve Step Programs for Eating Disorders”

I am not sure about the relevance of this section. It seems to describe whether meal plans are useful and gives the names of other programs. How do these relate to OA and how have they changed OA approaches/policy. This paper is about OA and descriptions of alternative programs and their history do not feel be useful as they are. I would recommend deleting unless it can be made relevant to OA. In any event the following should be clarified:

Line 243: provide a brief explanation of ““grey sheet” meal plan.”

Line 246: provide a brief description of “Dignity of Choice pamphlet”

Authors response: We thank you for expressing your receipt of this paragraph. The purpose of this paragraph was primarily to address common misconceptions that clinicians have of OA based on elements of other 12-step programs that are not affiliated with OA (ex: many dieticians incorrectly associate the “grey sheet diet” with OA, and therefore criticize OA for suggesting all members follow the same low-carbohydrate meal plan, when OA does not in fact endorse (but rather condones) the grey sheet diet. We have significantly restructured this section in order to make its purpose clearer; we have also provided brief explanations of the grey sheet meal plan and Dignity of Choice pamphlet (with citations to the actual contents of both). We hope this you will find these revisions satisfactory.

5.4. Is Overeaters Anonymous Synonymous with The Grey Sheet Diet? Variety in Twelve Step Eating Disorder Programs

Outside of OA, a variety of different twelve-step programs exist for eating disorders that are sometimes confused with OA. It is important for researchers, clinicians, and patients alike to understand which programs and program elements are- and are not associated with OA, as many clinical criticisms of OA pertain to programs or program aspects that are not actually OA-endorsed. For example, GreySheeters Anonymous (GSA) is a twelve-step fellowship of compulsive overeaters, separate from OA, who use the same “grey sheet” meal plan while working with a sponsor through OA’s twelves steps3. GSA and its namesake “gray sheet” diet/meal plan are often confused with OA as the “gray sheet meal plan” originated within OA in the late 1960s and was recommended by OA until it was replaced by the Dignity of Choice pamphlet4 in 19795. The “grey sheet meal plan” includes weight loss and maintenance options5,6; the meal plans’ restriction of carbohydrates opposes the U.S. Department of Agriculture’s recommended daily allowance of 130 g of carbohydrates per day with suggestions to “eat plenty of whole grains7.” In addition to risking nutrient deficiencies (ex: B vitamins, magnesium, and selenium that are found in whole grains)6, the carbohydrate restriction and overall restrictive nature of the meal plan6 have been criticized for risking carbohydrate demonization and creating unrealistic “food rules” that are associated with binge eating (see section 3.3). The Dignity of Choice book and pamphlet evolved from the grey sheet diet and offered 5 – 8 meal plan options to select from5,8. However, both the grey sheet diet and the Dignity of Choice meal plans oppose clinical beliefs that nutrition and meal planning must be unique to each individual (see section 3.3). As a result of these shortcomings, the OA World Service Business Conference decreed in 1997 that “offering food plans at OA meetings is a violation of Tradition 10… [and]… OA as a whole cannot print, endorse, or distribute food plan information to its members… [but] ought best concern [itself] with [its] suggested program of recovery – the Twelve Steps (Rescinded 2000)9,”(see Table 3 for Tradition 10)5.

Additional examples of twelve-step recovery groups for eating disorders that exist outside of OA include: Anorexics and Bulimics Anonymous (ABA); Compulsive Eaters Anonymous (CEA); CEA-Honesty, Openness, Willingness (CEA-HOW); Eating Disorders Anonymous (EDA); and Food Addicts Anonymous (FAA). All appear to have different food philosophy and culture. The importance of getting patients in touch with the right program and fellowship for them is described in section 3.3.

New comment:

This, again, reads like a defense of OA rather than a critical evaluation.  The point to me of this section in the context of a review on OA being an overlooked intervention, is to (more briefly) point out that OA may be overlooked as it is confused with other programmes.

We thank the reviewer for pointing out that this section read as ‘defense of OA.’ We have attempted to rewrite to be more objective. This section has been shortened considerably and focuses on three main points: 1) other 12-step programs exist that should not be considered synonymous with OA; 2) OA does not endorse any particular food plan; 3) offering food plans at OA meetings is considered a violation of OA’s 10th Tradition.

  • Old Comments:

Overeaters Anonymous: An Overlooked Intervention for Binge 2 Eating Disorder

Line 26 – “1-5% of the general population[2,3]” this is based on two references: [2] Kessler et al., 2013 and [3] Hutson et al. 2018

Kessler et al. state: “lifetime prevalence estimates average 1.0% for BN and 1.9% for BED across surveys” and “Country-specific lifetime prevalence estimates are consistently (median; interquartile range) higher for BED (1.4%; 0.8-1.9%)”

Hutson et al. state; “2-5% of the adult population and is more common in females than males (Dingemans et al., 2002, Kessler et al., 2013).” The Dingemans review was undertaken almost 20 years ago and at a time when BED is not a formal diagnosis within the then DSM-IV. Whilst estimating prevlance is difficult, it is important not to re-use figures from 2002 without explaining their limitations and to present the latest prevalence estimates clearly.

Authors response: We thank you for pointing this out. We have changed the statement to exclude the statistic from the Dingemans review. The statement now reads “with an estimated lifetime prevalence of 1.9% globally (Kessler et al) and 2.8% in the U.S. (Hudson et al). The “2.8% in the U.S.” is taken from Hudson et al.’s 2007 publication of “the prevalence and correlates of eating disorders in the National Comorbidity Survey replication.” In the survey and publication, the researchers analyzed data from the National Comorbidity Survey Replication (NCS-R). “NCS-R diagnoses were based on Version 3.0 of the World Health Organization Composite International Diagnostic Interview (CIDI) (Kessler and Ustun 2004), a fully structured lay administered diagnostic interview that generates diagnoses according to both ICD-10 and DSM-IV criteria” (Hudson et al., 2007). Notably, the CIDI criteria used for BED was in alignment with the current DSM-V diagnostic criteria for BED that requires the individual to have experienced at least 1 episode of binge eating weekly for at least 3 months of symptoms, and assesses for loss of control and marked distress through “a series of questions about attitudes and behaviors that are indicators of loss of control and of distress, rather than by direct questions,” (Hudson et al., 2007). Therefore, although the DSM-V diagnosis for binge eating disorder did not yet exist at the time of Hudson et al.’s 2007 publication, we feel that their diagnostic criteria was suitable to convey a relatively accurate measure of its prevalence at that time. 

This is addressed in 1) above.

Reviewer 3 Report

A very comprehensive albeit not systematic review on the topic. I am not sure of the aim of the article though as OA is utilised widely. Part of the OA/AA approach is to maintain anonymity an anonymous approach where participants can connect without repercussions and share freely, and further research may limit the actual benefits of the intervention due to this open approach.

Author Response

24 June 2021

Dear Reviewers –

We thank you for your thoughtful review and for your enthusiasm. We have attempted to address all of your comments and believe that your comments make this manuscript stronger and more relevant.

We have itemized the comments and our attempts to address them below. Please note that current responses are addressed in BLUE and previous responses are included in GREEN. Each reviewer’s responses are provided together, with the reviewer identified in DARK BLUE (ex: “Reviewer 1,” “Reviewer 2,” etc.).

Note that we have also added two sentences at the end of the introduction that bring current relevance to the field. These were not sentences suggested by the reviewers. However, binge-eating disorder is relevant to COVID lockdown. Thus, we felt this addition strengthened the manuscript.  

*

Reviewer 3

A very comprehensive albeit not systematic review on the topic. I am not sure of the aim of the article though as OA is utilized widely. Part of the OA/AA approach is to maintain anonymity an anonymous approach where participants can connect without repercussions and share freely, and further research may limit the actual benefits of the intervention due to this open approach.

Thank you for this valuable perspective. Although this manuscript provides a review of science, it was written as a communication that describes that strengths and limitations of OA. The purpose of the manuscript was not to increase use of OA, but rather provide a current description of the field.

We respect and value the opinion that further research could jeopardize OA’s tradition of anonymity and limit its actual benefits. Fortunately, this has not been the case for Alcoholics Anonymous (AA), from which OA evolved. Nevertheless, we have changed the verbiage of the final sentence of the abstract to state:

“Overall, OA provides a promising option for binge eating disorder treatment that warrants clinical research on its feasibility and efficacy in a way that respects and protects its tradition of anonymity.”

 We have also added the following sentence to the end of the last paragraph in section 2.2. (“Research on OA”): “It will be important for such trials to be conducted in a way that protects OA’s tradition of anonymity (Tool 4, Table 2; Tradition 12, Table 3), which enables members to share honestly and openly, without fearing gossip, thus deepening social connections and providing therapeutic benefit[5-10]. Although OA’s tradition of anonymity may pose a challenge for empirical research, it is one that research on other twelve-step interventions have overcome.”

We also added a final sentence to the end of the manuscript of a similar nature, which states: “It will be important for future research to be conducted in a way that protects OA’s tradition of anonymity and the therapeutic benefits it affords[5-10], as has been done in research on other twelve-step interventions."

Reviewer 4 Report

In this manuscript, the authors provide a comprehensive review of the intervention Overeaters Anonymous (OA), explain relevant concepts, and summarize findings on the effectiveness of OA as well as its clinical criticisms. Such a comprehensive review of OA is timely and provides a useful account of this intervention and the role of spirituality in the therapeutic effects. I have a few suggestions to improve the manuscript.

1), Abstract, lines 16-21, references required.

2), logical problems: lines 57-59, what is the association between feelings of depression and symptoms of depression? what is the association between data mentioned in lines 61-62 and that in lines 71-73? these can be better synthesized.

3), in section 3, the authors used OA to refer to both the intervention and the fellowship, which should be better differentiated to avoid confusion. For instance, consistent with the title of the manuscript, OA is better used to refer to the intervention only.

4), for the introduction of OA in section 3, the spiritual characteristics of the intervention should be explicitly mentioned. Relatedly, Section 5 may be better put before Section 4 because it provides more in-depth introduction to OA.

5), some citations of previous reports suffer from incomplete interpretation of the data, for instance, section 4.1 reference 90, the stated findings were true for subjects with low severity only. Furthermore, as for introduction to reference 113, what do the authors mean by "strongly correlated"? did they check the correlation coefficient?

6), the current review may be improved if the authors can specify the sample size and effect size measures when they introduce previous reports. For instance, findings of reference 108 in the 1st paragraph of section 4.3.3 are very well summarized, however, other parts are generally poorly written without mentioning sample size and effect size (although the authors did mention p value), e.g., section 4.3.2, 2nd paragraph on page 5.

7), section 4.3.4, please be reminded that not all meditations include spiritual connection, therefore, the statement here is problematic and should be revised. The discussion should be limit to meditations that involve spiritual connection only.

Author Response

24 June 2021

Dear Reviewers –

We thank you for your thoughtful review and for your enthusiasm. We have attempted to address all of your comments and believe that your comments make this manuscript stronger and more relevant.

We have itemized the comments and our attempts to address them below. Please note that current responses are addressed in BLUE and previous responses are included in GREEN. Each reviewer’s responses are provided together, with the reviewer identified in DARK BLUE (ex: “Reviewer 1,” “Reviewer 2,” etc.).

Note that we have also added two sentences at the end of the introduction that bring current relevance to the field. These were not sentences suggested by the reviewers. However, binge-eating disorder is relevant to COVID lockdown. Thus, we felt this addition strengthened the manuscript.  

Reviewer 4

In this manuscript, the authors provide a comprehensive review of the intervention Overeaters Anonymous (OA), explain relevant concepts, and summarize findings on the effectiveness of OA as well as its clinical criticisms. Such a comprehensive review of OA is timely and provides a useful account of this intervention and the role of spirituality in the therapeutic effects. I have a few suggestions to improve the manuscript.

1), Abstract, lines 16-21, references required.

Thank you for this comment. Reviewer 3 also suggested changes to the abstract. The abstract now reads as the following:

“The purpose of this communication is to provide an overview as well as strengths and weaknesses of Overeaters Anonymous (OA) as intervention for binge eating disorder treatment.”

We have been intentional in our wording such that references are not required. We feel that the new opening sentence also helps define the intended scope of the manuscript as a communication, rather than a review.

2), logical problems: lines 57-59, what is the association between feelings of depression and symptoms of depression? what is the association between data mentioned in lines 61-62 and that in lines 71-73? these can be better synthesized.

Thank you for this feedback. As per above, we have removed what was formerly the first paragraph of this introduction, so this feedback no longer applies.

Assuming we have correctly interpreted the 2nd part of the reviewer’s suggestions to pertain to lines 51-52 and 62-63, the purpose of this paragraph is to demonstrate the need for new interventions for BED. To clarify this, we added a sentence to the beginning of the paragraph stating, “There is a need for new interventions that can support binge eating disorder treatment.”

3), in section 3, the authors used OA to refer to both the intervention and the fellowship, which should be better differentiated to avoid confusion. For instance, consistent with the title of the manuscript, OA is better used to refer to the intervention only.

Thank you for this feedback. To clarify this, we have specified “the OA fellowship” when referring to the fellowship and have ensured that all use of only OA refers to the intervention only, in convention with the title, and as the reviewer has suggested. We added “The OA fellowship” to the beginning of the sentence stating: “The OA fellowship currently has >60,000 members…” We added “within the OA fellowship,” to the sentence stating: “Within the OA fellowship, most OA groups, formats, and sponsors suggest…”

4), for the introduction of OA in section 3, the spiritual characteristics of the intervention should be explicitly mentioned. Relatedly, Section 5 may be better put before Section 4 because it provides more in-depth introduction to OA.

Thank you. We did originally have the section on “Demystifying OA” (what you referred to as section 5) before the section “Are Twelve-Step Interventions Effective” (what you referred to as section 4); however, another reviewer suggested we swap the order. We have returned to the original order, as you suggested. We have also added in section 2 (introducing OA) the following sentence: “Although OA is non-affiliated, nondenominational, and non-religious, it is spiritual, as addressed in its literature[9-11], Steps 2 – 3[9-11] identified in Table 1 below, and in section 4.1 as well as the supplemental text provided in S1.”

5), some citations of previous reports suffer from incomplete interpretation of the data, for instance, section 4.1 reference 90, the stated findings were true for subjects with low severity only. Furthermore, as for introduction to reference 113, what do the authors mean by "strongly correlated"? did they check the correlation coefficient?

Thanks for catching this phrasing. We have added more precise contextual information regarding the MATCH study (formerly reference 90, now reference 71, lines 386 – 401).

Regarding the use of “strongly correlated” to introduce reference 113 (now reference 131) in line 449, the study provided Pearson Chi-square statistics, which can indicate association (not correlation), so we have changed “correlated” to “associated” in line 449 (and removed the word strongly), and added additional information on the findings as follows: “Specifically, relapse was found to be a function of strength of religious beliefs (x2 = 15.18, df = 3, p = 0.028; logistic regression = 10.65, df = 1, p = 0.006), frequency of attending religious services (x2 = 40.78, df = 5, p < 0.0005; logistic regression = 30.45, df = 1, p < 0.005), frequency of reading religious books (x2 = 27.190, df = 5, p < 0.0005; logistic regression = 17.31, df = 1, p < 0.0005), frequency of watching religious programs (x2 = 19.02, df = 5, p = 0.002; logistic regression = ns), and frequency of meditation and/or prayer (x2 = 11.33, df = 5, p = 0.045; logistic regression = 9.650, df = 1, p = 0.002), with spiritual participants reporting 7%–21% less drug and alcohol use than non-spiritual subjects (excluding crack).”

We have also added additional information to the citations that will be useful for interpretation (ex: n, p-value, etc.).

6), the current review may be improved if the authors can specify the sample size and effect size measures when they introduce previous reports. For instance, findings of reference 108 in the 1st paragraph of section 4.3.3 are very well summarized, however, other parts are generally poorly written without mentioning sample size and effect size (although the authors did mention p value), e.g., section 4.3.2, 2nd paragraph on page 5.

Thank you. We have done this.

7), section 4.3.4, please be reminded that not all meditations include spiritual connection, therefore, the statement here is problematic and should be revised. The discussion should be limit to meditations that involve spiritual connection only.

Thank you. In response to another reviewer’s feedback, we have removed this section from the manuscript.

References:

  1. Kriz, K.-L.M. The Efficacy of Overeaters Anonymous in Fostering Abstinence in Binge-Eating Disorder and Bulimia Nervosa. Dissertation, Virginia Polytechnic Institute and State University, Falls Church, Virginia, 2002.
  2. Russell-Mayhew, S.; von Ranson, K.M.; Masson, P.C. How does overeaters anonymous help its members? A qualitative analysis. European eating disorders review : the journal of the Eating Disorders Association 2010, 18, 33-42, doi:10.1002/erv.966.
  3. Martin, D.D. From Appearance Tales to Oppression Tales:Frame Alignment and Organizational Identity. Journal of Contemporary Ethnography 2002, 31, 158-206, doi:10.1177/0891241602031002003.
  4. von Ranson, K.; Russell-Mayhew, S.; Masson, P. An exploratory study of eating disorder psychopathology among Overeaters Anonymous members. Eating and weight disorders : EWD 2011, 16, e65-68, doi:10.1007/BF03327524.
  5. OA, O.A. Working the Program. Available online: https://oa.org/working-the-program/ (accessed on Nov 4, 2020).
  6. OA, O.A. The Nine Tools of OA. Available online: https://www.oaregion10.org/newcomers/the-nine-tools-of-oa/ (accessed on Nov 4, 2020).
  7. VSB, V.S.B.o.O.-H. Sponsee Guidelines. For Sponsors & Sposnees: Sponsee Guidelines 2018, 1.
  8. OA, O.A. Two-Hour OA HOW Phone Meetings. Available online: (accessed on Nov 4, 2020).
  9. OA, O.A. The Twelve Steps and Twelve Traditions of Overeaters Anonymous, 2 ed.; Overeaters Anonymous, Inc. World Service Office: Rio Rancho, NM, 2018.
  10. AA, A.A. Twelve steps and twelve traditions: A co-founder of Alcoholics Anonymous tells how members recover and how the society functions; Alcholics Anonymous World Services, Inc.: New York, NY, 1981.
  11. AA, A.A. Alcoholics Anonymous: The story of how many thousands of men and women have recovered from alcoholism, 4 ed.; Alcoholics Anonymous World Services, Inc.: New York, NY, 2001.
  12. Schoenthaler, S.J.; Blum, K.; Braverman, E.R.; Giordano, J.; Thompson, B.; Oscar-Berman, M.; Badgaiyan, R.D.; Madigan, M.A.; Dushaj, K.; Li, M.; et al. NIDA-Drug Addiction Treatment Outcome Study (DATOS) Relapse as a Function of Spirituality/Religiosity. J Reward Defic Syndr 2015, 1, 36-45, doi:10.17756/jrds.2015-007.
  13. Yeterian, J.D.; Bursik, K.; Kelly, J.F. "God put weed here for us to smoke": A mixed-methods study of religion and spirituality among adolescents with cannabis use disorders. Subst Abus 2018, 39, 484-492, doi:10.1080/08897077.2018.1449168.

Round 2

Reviewer 1 Report

The Authors have deleted their review of neuroscience from the manuscript and expanded on the non-scientific topics of OA in e.g., ”Research on Overeaters Anomymous (line 261)”

They say that ”Research on the effectiveness of OA is also supportive” by referring to websites, a 2002 PhD dissertation, case reports and mentalistic stuff related to ”spirituality”, ”God” etc rather than published papers. This is outside the bounds of science.  

As the authors point out ”randomized controlled trials are needed in order to empirically determine OA’s effectiveness for clinical use” and before anything approaching proper testing of the ideas of OA has been undertaken little, if anything, of what the authors say makes sense. And ”4.3. Research on Spirituality …” (line 321) makes no sense at all.

But suppose that the authors are right in that Sprituality” and ”God” may be useful in treating BED. Reference 25 provides some support. This support is, however, meagre by all measures. And so before OA for BED is taken into serious consideration its theoretical/scientific basis needs to be stated and its hypotheses need to be formulated and tested. The authors´ manuscript does not approach minimal scientific standards.

This Reviewer finds it impossible to make sense of the authors text and so stopped reviewing after some of the Sprituality/God stuff and did not check if the chocolate milk shake is still included in the manuscript.   

Reviewer 4 Report

Thank the authors for addressing my concerns.

Author Response

Pleasure. Thanks again for your invaluable feedback!

This manuscript is a resubmission of an earlier submission. The following is a list of the peer review reports and author responses from that submission.

Round 1

Reviewer 1 Report

Thank you for the opportunity to review this manuscript on OA.  I have made some comments and suggestions that I hope will improve the scientific validity and clarity of the script, which I attach.

Reviewer 2 Report

This Reviewer enjoyed reading this manuscript in which the authors correctly note that BED is the most common eating disorder and that treatment outcomes can be improved as remission rates are just over 50% and relapse is a major concern. They aim at launching a 12 step program for improving outcomes, a laudable initiative.

Abstract -  Please let readers know what makes this program promising and please dispense with concepts such as ”God” and ”Higher Power” (at least in the Abstract), as IJERPH is a scientific journal. Also, please state the results that you have obtained rather than saying that you have ”recognized common criticisms” and that the program is ”promising”.

Introduction – 1.1., 1.2. and 1.2.1. are clearly written and supported by a comprehensive list of references. How can 42-96% of adults with BED meet DSM criteria for substanse-related and addictive disorder yet only 27% have BED and comorbid substanse-related and addictive disorder (lines 68-74)?

1.2.2. The topics of homeostasis and non-homeostatis have been reviewed many times before and the authors add nothing  new, their suggestion that Berridge´s systems of ”liking” and ”wanting” are ”pure ´go´ systems” that do not generate strong ”stop” signals etc are re-wordings of what has been said in the papers and reviews that the authors list. The same is true for what the authors list on neurochemistry and imaging studies, their overview is a repeat of previous literature.

It is important, however, that ”These systems are thought to be inherent to all reward processes” (lines 88-89), although ”reward” might be better replaced by ”reinforced” because ”reward” is indicative of something enjoyable and even ”pleasant” and these should be kept separate1.

GWAS-studies are hypothesis-free and cannot speak of causality2 and are at best interesting before mechanistic hypotheses have been proposed. In the case of weight regulation and eating behavior, GWAS-studies are unlikely to provide anything that can be used to control body weight pharmacologically, which is their aim. This is unsurprising and now generally recognized because of the many genes involved. This is also the case for many, if not most, medical problems, including CVD3,4. This is too well know to require supporting references. The failure of genetic analysis in improving the human condition is also apparent for non-medical problems, e.g., how to improve on athletic performance5.   

The authors analysis thus repeats what is known already, nothing of which has generated anything that can be used to treat under- or overweight patients.

In 2000 the conventional homeostasis model was revised by the demonstration that a then popular orexigen, neuropeptide tyrosine (NPY), was found to decrease food intake when the search for food was experimentally circumvented6. This observation was extended by the demonstration that NPY facilitates anorexia in an animal model of Anorexia Nervosa7, and the findings were subsequently translated into the clinic8. This new role of the neuroendocrine cells in the medial basal hypothalamus was subsequently confirmed 9–11, an update appeared recently12. The brain has evolved to assist an individual in the search for food, satiety has not been encouraged in evolution3,4,13.

These (now not so) new findings need to be taken into consideration. Particularly so as the Bernard-Cannon idea of ”homeostasis” emerged from the study of depletion, not abundance8 and also because the causes of the present problems of obesity were outlined in 1953 already14.  

Thus, the authors embark on their ”Overeaters Anonymus-project” (OA) on the background of an outdated view of brain function. And in a minute they transit from eating and the hypothalamus to ”God” and ”Higher Powers”! This jump should have been avoided by reviewing the neuroscience of ”God” and ”Higher Powers” rather than the neruoscience of eating and body weight regulation, a less well researched topic although some information is available15.

The authors suggest (”conclude”?) that ”clinics, clinicians, and individuals can use OA to achieve, maintain, enhance, and support recovery from a variety of eating disorders, including binge eating disorder” (lines 163-165) and list 14 supporting references (19, 49, 61,62, 63, 82, 83, 84, 90, 91, 92, 93, 94, and 95). They also say that: ”Numerous studies show that twelve-step interventions are effective in promoting short- and long-term resilience and recovery from substance-related and addictive disorders, both when used as a component of a larger treatment model and/or when sought” (lines 259-262). But they then say that: ”Research on the efficacy of OA is limited but supportive” (line 277).   

One of the tools of OA that the authors list: ”Tool 1, Plan of eating”, is similar to the first step i CBT for binge eating, i.e., the establishement of a regular pattern of eating, which is the most important part of CBT for BED and also other eating disorders.16 The other tools aim at motivating subjects to comply with Tool 1, none of which has been demonstrated effective in a scientific analysis.

Before the OA program used in  the clinic it should be demonstrated effective in an RCT.   

1. Berridge KC, Kringelbach ML. Pleasure systems in the brain. Neuron. 2015;86(3):646-664. doi:10.1016/j.neuron.2015.02.018
2. Pearl J, Mackenzie D. The Book of Why: The New Science of Cause and Effect. 01 edition. Penguin; 2019.
3. Lieberman DE. Evolution of the Human Head. 1st Edition edition. Harvard University Press; 2010.
4. Lieberman D. Exercised: The Science of Physical Activity, Rest and Health. Allen Lane; 2020.
5. Epstein D. The Sports Gene: Talent, Practice and the Truth About Success. Yellow Jersey; 2014.
6. Södersten P, Nergårdh R, Bergh C, Zandian M, Scheurink A. Behavioral neuroendocrinology and treatment of anorexia nervosa. Front Neuroendocrinol. 2008;29(4):445-462. doi:10.1016/j.yfrne.2008.06.001
7. Dietrich MO, Zimmer MR, Bober J, Horvath TL. Hypothalamic Agrp neurons drive stereotypic behaviors beyond feeding. Cell. 2015;160(6):1222-1232. doi:10.1016/j.cell.2015.02.024
8. Chen Y, Lin Y-C, Kuo T-W, Knight ZA. Sensory detection of food rapidly modulates arcuate feeding circuits. Cell. 2015;160(5):829-841. doi:10.1016/j.cell.2015.01.033
9. Burnett CJ, Li C, Webber E, et al. Hunger-Driven Motivational State Competition. Neuron. 2016;92(1):187-201. doi:10.1016/j.neuron.2016.08.032
10. Wang C, Zhou W, He Y, et al. AgRP neurons trigger long-term potentiation and facilitate food seeking. Transl Psychiatry. 2021;11. doi:10.1038/s41398-020-01161-1
11. Södersten P, Bergh C, Zandian M, Ioakimidis I. Obesity and the brain. Med Hypotheses. 2011;77(3):371-373. doi:10.1016/j.mehy.2011.05.018
12. Mayer J. Genetic, traumatic and environmental factors in the etiology of obesity. Physiol Rev. 1953;33(4):472-508. doi:10.1152/physrev.1953.33.4.472
13. Borg J, Andrée B, Soderstrom H, Farde L. The serotonin system and spiritual experiences. Am J Psychiatry. 2003;160(11):1965-1969. doi:10.1176/appi.ajp.160.11.1965
14. Fairburn CG, Marcus MD, Wilson GT. Binge Eating, Nature, Assessment, and Treatment. Guilford Press; 1993.
15. Dalle Grave R, Calugi S. Cognitive Behavior Therapy for Adolescents with Eating Disorders. The Guilford Press; 2020.